# Analysis of Big Game Fishing Catches of Blue Marlin (*Makaira nigricans*) in the Madeira Archipelago (Eastern Atlantic) and Factors that Affect Its Presence

Roi Martinez-Escauriaza [1,*], Pablo Pita [2,3], Maria Lídia Ferreira de Gouveia [4], Nuno Manuel Abreu Gouveia [5], Eduardo Teixeira [6], Mafalda de Freitas [1,4,7,8] and Margarida Hermida [1,8]

1   Oceanic Observatory of Madeira, Agência Regional para o Desenvolvimento da Investigação Tecnologia e Inovação (ARDITI), Edifício Madeira Tecnopolo, 9020-105 Funchal, Portugal; mafalda.freitas.araujo@madeira.gov.pt
2   Campus Do Mar, International Campus of Excellence, 15782 Santiago de Compostela, Spain; pablo.pita@usc.es
3   Faculty of Political and Social Sciences, University of Santiago de Compostela, 15782 Santiago de Compostela, Spain
4   Direção Regional do Mar, Direção de Serviços de Monitorização, Estudos e Investigação do Mar (DRM/DSEIMar), 9004-562 Funchal, Portugal; lidia.gouveia@madeira.gov.pt
5   Direção Regional de Pescas, Direção de Serviços de Inspeção e Controlo, Edifício da Sociedade Metropolitana de Câmara de Lobos, 9300-138 Câmara de Lobos, Portugal; nuno.gouveia@madeira.gov.pt
6   Big Game Club of Portugal in Madeira, 9000-171 Funchal, Portugal; vetfunchal@mail.telepac.pt
7   Estação de Biologia Marinha do Funchal, Cais do Carvão, 9000-003 Funchal, Portugal
8   MARE–Marine and Environmental Sciences Centre, Agência Regional para o Desenvolvimento da Investigação Tecnologia e Inovação (ARDITI), Edifício Madeira Tecnopolo, 9020-105 Funchal, Portugal; margarida.hermida@mare-centre.pt
*   Correspondence: roimartinez@hotmail.com

**Abstract:** The archipelago of Madeira (Portugal) is one of the main European big game fishing locations, where the main target species is the blue marlin (*Makaira nigricans*). Catch data for these fish were used to analyze their presence over the years, estimate their average weights, and calculate annual fishing success rates. The results showed a marked seasonal effect, with higher average catch rates in summer (June–July), suggesting a migration from the equatorial waters they inhabit at the beginning of the year to northern areas when the waters become warmer. The influences of some environmental factors were analyzed using generalized additive models, and it was observed that the occurrence of blue marlin may be influenced by water temperature, wind, rain, and atmospheric pressure. This fishery did not register a high mortality rate in blue marlin specimens due to the usual practice of catch and release; individuals captured in this fishery can be used as a source of information that allows for follow-up on the status of the blue marlin population in the region.

**Keywords:** sport fishing; Macaronesia; GAM; Portugal; catch and release; seasonality

## 1. Introduction

Blue marlin (*Makaira nigricans*, Lacepède, 1802) is an epipelagic oceanic species distributed throughout tropical regions, which spends most of its time near the surface at night and at greater depths (25–100 m) during the day [1]. The blue marlin is the largest species of the family Istiophoridae, reaching weights of up to 625 kg [2], with a tropical and temperate distribution throughout the world [3]. Its latitudinal range in the Atlantic extends from about 45° N to about 35° S [4]. Its total distribution is generally associated with sea surface temperature (SST) isotherms >24 °C [5]. Even so, in the Atlantic Ocean, adults are commonly found in waters with SST ranging between 22 and 31 °C [6,7]. Although this species has a low market value, catches are not negligible, especially due to the frequent accidental catches by pelagic longliners targeting tuna (*Thunnus* spp.) and

swordfish (*Xiphias gladius* Linnaeus, 1758). That said, small catches are also made in coastal fisheries that use gillnets, harpoons, and purseseines [8]. This phenomenon of accidental bycatch has led to a large decline in blue marlin stocks [9]. Standardized catch per unit effort (CPUE) indices of abundance for blue marlin depict a monotonically decreasing trend from the early 1960s to the early 2000s [10]. In the Atlantic, billfish landings represent only 0.76 percent of the combined tunas, swordfish, and billfish species which makes the collection of billfish stock assessment data through formal fishery statistical systems challenging [10]. Landings of Atlantic blue marlin fluctuated between 3000 and 4000 tonnes during the 2000s, most of them originating from longline operations and gillnetting [10]. Food and Agriculture Organization (FAO, Rome, Italy) describes significant uncertainties in the state of their exploitation that represent a serious concern. In the Atlantic, blue marlins seem to be overexploited even though they are not generally targeted, while in the eastern Pacific they are fully exploited [11]. The scarcity of available information has motivated research with the goal of better understanding the biology and conservation of this species [4,9].

In addition to being a frequent bycatch species in commercial fisheries, blue marlin is the target of a worldwide recreational fishing industry based on the capture of large pelagics [10,12]. This activity is known as "big game fishing," in which fishers troll with live bait or with jigs to catch a specimen and, in case of hooking, the objective is to draw the catch close to the boat for subsequent release [2]. Blue marlin, which performs spectacular acrobatics when caught on rod and reel, is a very attractive catch for recreational fishers, making it one of the most important game fish species in the world [2,4]. The catch-and-release (C&R) approach to marlin fishery does not allow for the collection of extensive information about the species. A further consequence of this practice is that fish sometimes require resuscitation prior to release or may die during the struggle [12,13]. Anglers' experience differs greatly depending on their handling skill level and C&R behaviors, which influence the short- and long-term physiological consequences for angled fish and, in turn, determine their survival outcomes [14]. Furthermore, marlins are sometimes landed for measuring when there is a possibility of breaking a record, a situation that disproportionately affects larger specimens.

Other billfishes, mainly the white marlin (*Tetrapturus albidus*; Poey, 1860), are also sometimes captured. Additionally, some other species, such as wahoos (*Acanthocybium solandri*, Cuvier, 1831), dolphinfishes (*Coryphaena hippurus*, Linnaeus, 1758 and *Coryphaena equiselis*, Linnaeus, 1758), and various species of tuna (mainly bigeye tuna—*Thunnus obesus*, Lowe, 1839; albacore—*Thunnus alalunga*, Bonnaterre, 1788, and skipjack—*Katsuwonus pelamis*, Linnaeus, 1758), are landed for their high gastronomic value.

In the Macaronesian region (eastern Atlantic), some fishing companies and many amateurs practice this type of fishing [15,16]. In Madeira, the good weather and the proximity of fishing grounds attract many fishers during the high season, which usually lasts from May to September [5,17,18]. Most vessels fish in this specific season, and, when it is over, the boats are usually taken to dry dock for repairs until the following year. Due to the year-long good weather conditions, some companies extend their activity over almost the entire year, targeting other large fish when marlins are scarce. The presence of blue marlin varies seasonally, and their density is influenced by the interannual variability of oceanographic and environmental factors [8,19].

There are significant knowledge gaps regarding this type of fishing, given the effort required, the seasonality, the fleet, and the C&R approach. The aim of the present study was to investigate the state of big game fishing in Madeira and how it affects the population of blue marlin. The specific objectives were to analyze blue marlin captures in the region to know their seasonal and annual variation, the average weight of the individuals in the region and the dynamics of blue marlin in Macaronesian archipelagos, and to evaluate how environmental factors can affect the presence of blue marlin.

## 2. Materials and Methods

### 2.1. Data Acquisition

In order to assess the size of the big game fishing fleet, as well as the number of targeted species in Madeira, a list was compiled of all the companies in the region dedicated to this fishing modality. A list of boats was obtained after consulting with local captains, whose vessels were docked in Funchal (southern coast of Madeira) or based in the Calheta marina (southwest coast of Madeira) (Figure 1).

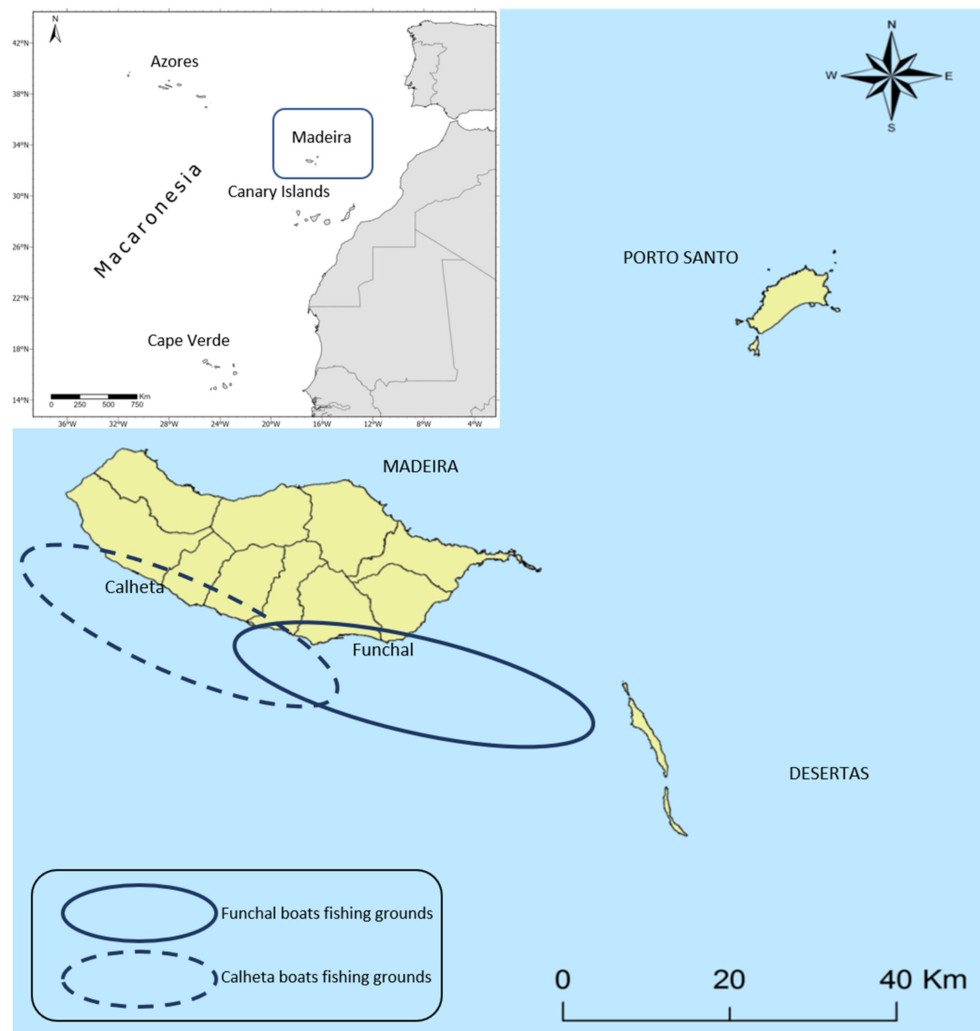

**Figure 1.** Madeira archipelago location, marinas harboring big game fishing boats, and their fishing grounds.

### 2.2. Big Game Fishing in Madeira

Historical data on blue marlin caught from 2008 to 2019 by the Madeira big game fishing fleet were collected through contact with captains, and also from the billfishreport. com database (accessed on 20 May 2020). This webpage provides daily blue marlin catch reports with pictures of specimens, which are provided by charter and private boats practicing big game fishing across the world. The crews must register on the page and record their catches, with location, boat name, captain, and specimen weight included. Registers obtained from both sources indicated seasonality, number of marlins captured, and average weight of individuals. Records of catches obtained across multiple years can give an approximation of the number of blue marlins captured each year, thus allowing the interannual fluctuations to be estimated.

Conversations with crews and observation of some fishing trips revealed that the main fishing grounds are located on the south coast of the island, protected from the northeast trade winds by high cliffs. The boats from Funchal normally cover an area slightly to the east, and sometimes close to the Desertas Islands. The boats from Calheta usually fish near the southwest of the island (Figure 1).

From the period 2017 to 2019, a more detailed register was collected using several different sources, including questionnaires directed to captains, logbooks, and online reports of catches (billfishreport.com, accessed on 20 May 2020)). From this information, the number of fishing trips and catches, species identification, and estimated weights (visually estimated by the captains according to the specimen size since they were not usually captured) were registered and compiled in a database.

In 2017, the big game fishing fleet in Madeira was composed of 31 vessels, including 20 charters and 11 private boats. They were distributed across two marinas: Funchal, with 9 charters and 3 private, and Calheta, with 11 charters and 8 private. In 2018, 3 additional boats were active:1 private in Funchal, a charter in Calheta, and a third private vessel fishing in both marinas. In 2019, 3 vessels from Calheta ceased their activity (Table 1). For the analysis, both vessel categories were considered together.

**Table 1.** Number of boats for each category in each marina (2017–2019).

|  |  | **2017** | **2018** | **2019** |
|---|---|---|---|---|
| Funchal | Private | 3 | 5 | 5 |
|  | Charter | 9 | 9 | 9 |
| Calheta | Private | 8 | 8 | 8 |
|  | Charter | 11 | 12 | 9 |

The information for posterior analyses was obtained from a total of 18 vessels in 2017. In 2018, 4 boats were added to the analysis to give 22 in total, and in 2019 only 19 boats provided this information.

The data collected were used to calculate fishing effort using the number of hours trolled and the number of fishes caught for all species and blue marlin specifically. The results allowed the calculation of catch per unit effort (CPUE) and comparison between boats and years. CPUE is calculated by dividing the number of fish caught by the total fishing effort [20]. The CPUE for blue marlin was also calculated separately. Fishing effort was defined as the total number of boats multiplied by the total number of hours trolled, including trips that did not result in a catch. Unfortunately, it was impossible to determine the effort made by some of the boats.

Registers of other species captured by the big game fishing fleet and the percentage of the total capture were also analyzed.

The average weight of blue marlin individuals captured was also studied, and records of dead specimens were analyzed in order to ascertain the impact of this fishery on fish populations. In addition, via the billfishreport.com webpage (accessed on 20 May 2020), we accessed the register of blue marlin captured in different regions of the world, along with their estimated weights, from 2011 to 2019. This information was used to analyze the numbers of specimens, the seasonality, and the weights registered for the Macaronesian archipelagos of Azores, Madeira, Canary Islands, and Cape Verde, to make comparisons between them, and to propose how these fish migrate within this geographic region throughout the year.

### 2.3. Relationship between CPUE of Blue Marlin and Environmental Variables

The hours trolled and numbers of blue marlin caught monthly from 2017 to 2019 were analyzed to determine any relation to environmental factors. The average monthly sea surface temperature (SST; °C) for the fishing ground was obtained from NOAA (www.esrl.noaa.gov/psd/, accessed on 13 September 2020). Atmospheric pressure (hPa),

rain (mm), cloud cover (%), and wind speed (km/h) were obtained from World Weather Online (www.worldweatheronline.com, accessed on 13 September 2020), and the monthly value of the North Atlantic Oscillation Index (NAO) was obtained from NOAA (www.ncdc.noaa.gov/teleconnections/nao/, accessed on 13 September 2020).

Multiple regression models were used to individually study the variations in the catches of blue marlin (between 2017 and 2019) in relation to variations in their environment. Dates of observation (month and year), average monthly SST, rain, pressure, wind, and NAO were included in the models as explanatory continuous variables. The monthly CPUE was used to carry out these analyses because it was considered the most appropriate variable for analyzing long-term trends in fish availability and the associated factors.

Specifically, generalized additive models (GAM) [21,22] were used to analyze the relationships between environmental factors and the monthly CPUE of blue marlin in the area. Unadjusted models were obtained first, and a forward stepwise selection procedure was followed to fit the adjusted models (i.e., from unadjusted to saturated models). The models were fitted, selected, and validated via the mgcv package [23] in R, ver. 4.0.2 [24]. Smoothed functions were used for the quantitative response variables, with penalized thin-plate regression splines [25]. Different error structures and link functions were assessed in the models. The final models were selected based on Akaike's criterion (AIC) [26], with the percentage of explained variance as a secondary criterion. The gam.check tool of the mgcv package, which plots deviance residuals against the approximate theoretical quartiles of the deviance residual distribution according to the fitted model, was used to check the residuals of the models [22]. Models with over dispersed and anomalous residual distribution were discarded. Subsequently, the predict tool of the mgcv package was used to infer the response variable, using the best performing models to show the results. For these estimates, the mean value of the quantitative explanatory variables was used in each case.

## 3. Results

### 3.1. Effort and CPUE

Using the registers of fishing trips made from 2017 to 2019 by the big game fishing fleet boats that provide this information, it was possible to calculate and compare the hours spent on fishing activity and the catches made. These data indicate the effort made by each boat, and allowed the calculation of their CPUEs (one directed at all the species captured and the other focused only on blue marlin catches) (Table 2).

**Table 2.** Comparison between the two marinas (Calheta and Funchal) and two boat categories (private and charter) of fishing trips, hours and CPUE (average), and total values of the vessels from 2017 to 2019 (BM: blue marlin).

| | 2017 | | | | | 2018 | | | | | 2019 | | | | |
|---|---|---|---|---|---|---|---|---|---|---|---|---|---|---|---|
| | Trips | | | CPUE | BM CPUE | Trips | | | CPUE | BM CPUE | Trips | | | CPUE | BM CPUE |
| | Numbers | Hours | Total (h) | | | Numbers | Hours | Total (h) | | | Numbers | Hours | Total (h) | | |
| Calheta | 61.7 | 7.1 | 440.9 | 0.057 | 0.033 | 44.6 | 6.9 | 314.9 | 0.033 | 0.015 | 52.8 | 7.2 | 381.8 | 0.08 | 0.048 |
| Funchal | 75.3 | 5.5 | 416.6 | 0.034 | 0.018 | 53.1 | 6 | 321.8 | 0.08 | 0.014 | 39.3 | 6.2 | 235.2 | 0.062 | 0.051 |
| Charter | 67 | 6 | 398.9 | 0.051 | 0.027 | 53 | 6.4 | 331.5 | 0.069 | 0.015 | 48.9 | 6.4 | 314.8 | 0.082 | 0.056 |
| Private | 69.6 | 8 | 556.8 | 0.044 | 0.028 | 39.9 | 7.1 | 306.8 | 0.026 | 0.017 | 47.8 | 7.8 | 375.8 | 0.053 | 0.036 |
| Total | 1192 | 6.5 | 7790.4 | 0.048 | 0.027 | 1048 | 7 | 6983.5 | 0.04 | 0.015 | 857 | 7 | 5760 | 0.07 | 0.05 |

Over the course of 1192 registered trips and a total of 7790 h of fishing in 2017, 216 blue marlins were captured, with a CPUE for blue marlin of 0.028. In 2018, a total of 1048 fishing trips were carried out, with a total fishing time of 6983 h and 104 blue marlins caught, resulting in a CPUE of 0.015. In 2019, a total of 280 blue marlins were caught, corresponding to a CPUE of 0.049 and a total of 5760 h trolled over 857 fishing trips.

Unfortunately, it was not possible to obtain data concerning the number of trips made by some of the boats during this period, and therefore it was not possible to calculate the total effort for the entire fleet.

### 3.2. Other Species

Besides blue marlin, some other pelagic species were caught with the trolling technique examined in this study (Table 3). The percentage of catches of other species in relation to blue marlin was calculated, and the results showed a considerable variability according to the year. In 2017, 42.6% of the total catches were other species; this value was 59.7% the following year (2018), and in 2019 only 30% of fishes caught belonged to species other than blue marlin.

**Table 3.** Number of individuals of other species captured by the big game fishing fleet.

|  | 2017 | 2018 | 2019 |
|---|---|---|---|
| White marlin | 18 | 8 | 10 |
| Big eye tuna | 40 | 26 | 18 |
| Wahoo | 32 | 8 | 17 |
| Dolphinfish | 14 | 111 | 35 |

### 3.3. Seasonal and Annual Variation in Blue Marlin Catches

The blue marlin fishing season in Madeira takes place from April to November. According to historical data, blue marlin was the most captured species by the big game fishing fleet in Madeira over the last 10 years. Blue marlins appear between May and September; only a couple of specimens have been captured in October. June and July can be considered the peak fishing season for this species (Figure 2).

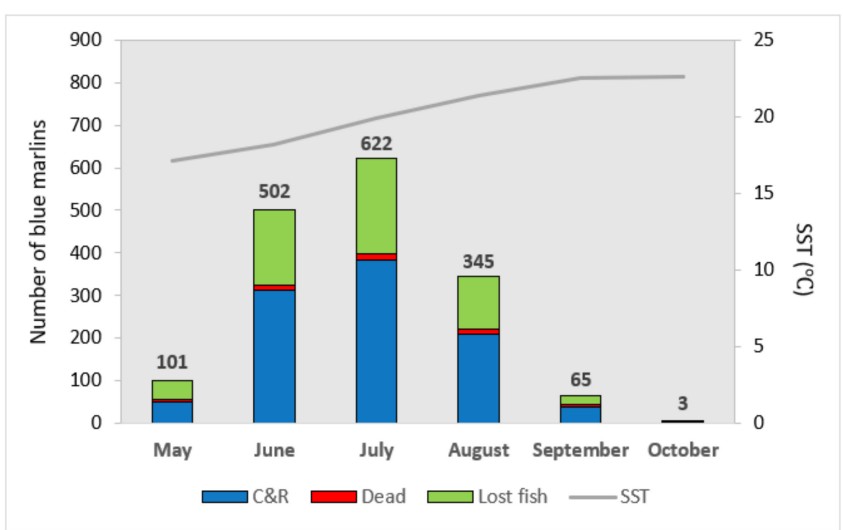

**Figure 2.** Number of registered blue marlin individuals by month and mean monthly SST from 2012 to 2019 (lost fish: fish that escaped after biting the lure; C&R: catch and release; SST: sea surface temperature).

With regard to the interannual fluctuation in blue marlin catches, the records show a high presence in 2008 and 2009, with the highest number (649) recorded in 2009. After that, there was a significant reduction, especially in 2011 and 2013, but from 2014 onwards a slight increase was observed, with 2019 reaching levels close to those of 2008 (Figure 3).

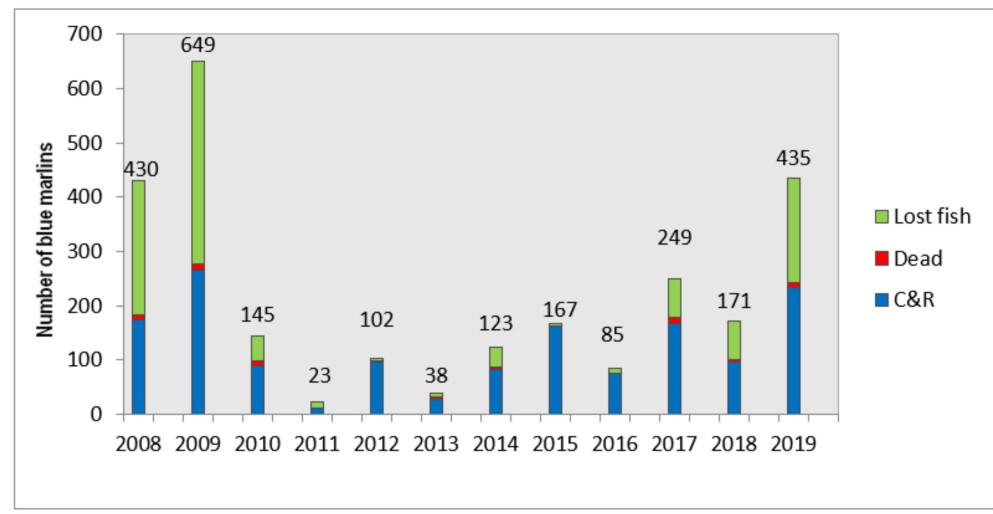

**Figure 3.** Number of registered blue marlin individuals catches by year from 2008 to 2019 (lost fish: fish that escaped after biting the lure; C&R: catch and release).

Additionally, the number of blue marlins captured per month was obtained for each Macaronesian archipelago using the registers of catches on billfishreport.com (accessed on 20 May 2020). These data allowed us to plot a graph showing the seasonal occurrence of blue marlin in different areas (Figure 4). The first catches were obtained in Cape Verde in February, although they were not abundant until March. With the arrival of spring, they become more abundant, whereas in summer the catches began to decrease. In late spring and early summer, the blue marlins became more prevalent in the Canary Islands, moving north and reaching their peak in Madeira between June and July. They then continued to migrate to latitudes further north, and in August and September, the greatest abundance was registered in the Azores.

*3.4. Estimated Weight Analysis*

The blue marlin weights estimated by captains are shown in Table 4. Unfortunately, no historical data regarding weights exist from 2008 or 2009, the years when most blue marlins were caught. Between 2011 and 2019, weights ranged from 45.4 (2018) to 499 (2015) kg. The average weight of blue marlin in Madeira was 290.4 kg.

**Table 4.** Weights in kg (average, SD—standard deviation, *n*—number of individuals, Min—minimum and Max—maximum) of blue marlin from 2011 to 2019.

| Year | Average Weight (kg) | SD | *n* | Min | Max |
|------|---------------------|------|-----|-------|-------|
| 2011 | 322.6 | 61.9 | 9 | 226.8 | 385.6 |
| 2012 | 274.8 | 79.1 | 47 | 100 | 430.9 |
| 2013 | 285.1 | 98.8 | 130 | 112.9 | 476.3 |
| 2014 | 295.8 | 76.9 | 53 | 99.8 | 453.6 |
| 2015 | 319.5 | 74.8 | 130 | 113.4 | 499 |
| 2016 | 301.7 | 72.5 | 59 | 63.5 | 445.9 |
| 2017 | 285.3 | 75 | 168 | 68 | 437.7 |
| 2018 | 287.5 | 70.1 | 84 | 45.4 | 408.2 |
| 2019 | 273.6 | 80.6 | 206 | 90.7 | 453.6 |
| 2011–2019 | 290.4 | 77.3 | 886 | 45.4 | 499 |

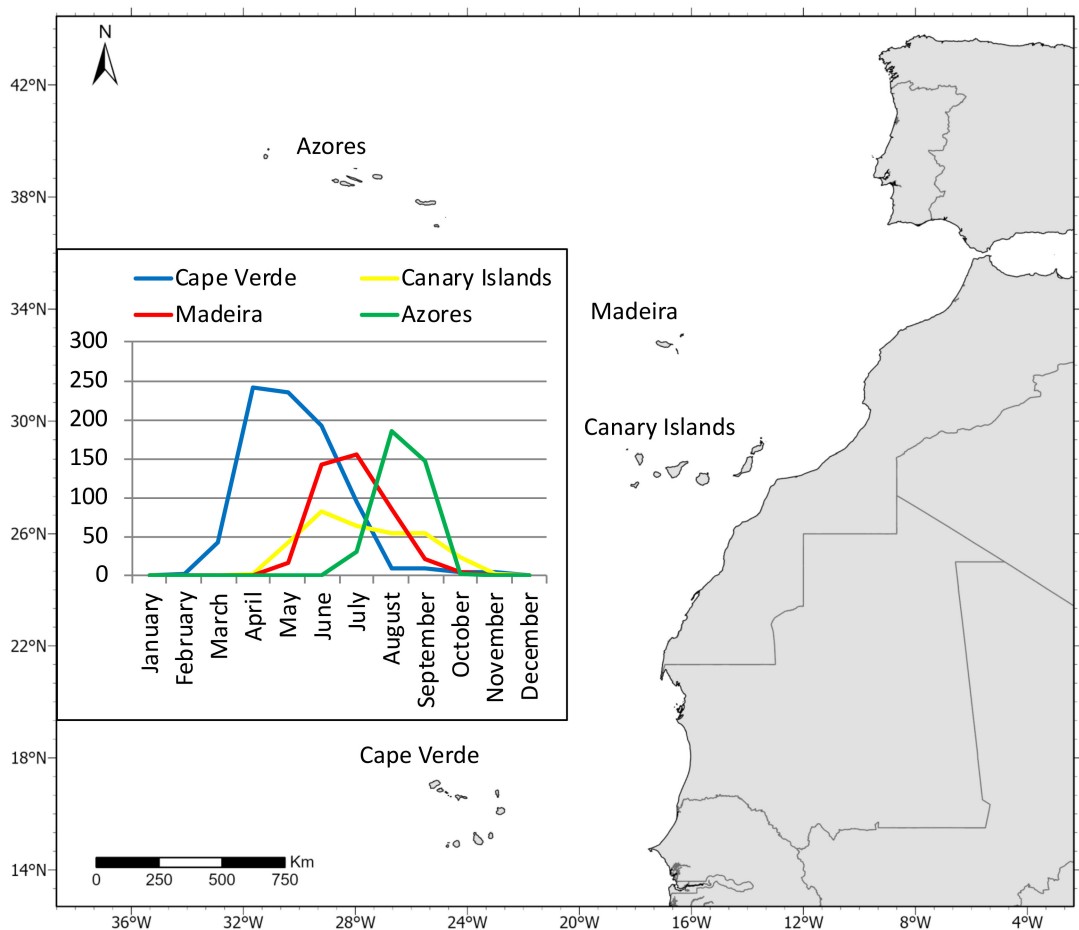

**Figure 4.** Macaronesian archipelagos and the presence of blue marlin throughout the year (2011 to 2019).

In the last 11 years (2008–2019), a total of 67 blue marlins were landed, corresponding to 4.2% of the total captured. From 2008 to 2010, around 10 blue marlins were landed each year. Not many blue marlins died in the following years, except for in 2017 and 2019, in which 11 and 9 blue marlins, respectively, died and were landed. Considering the weights of dead specimens registered, it was possible to calculate the overall weight of the individuals captured in recent years. In 2017, 2845.7 kg of blue marlin was landed in Madeira by recreational fisheries. In 2018, the five dead blue marlins weighed a total of 1131.3 kg, while in 2019, a total of 2437.6 kg of blue marlin was landed, as inferred from the nine dead specimens. In addition, the Regional Directorate of Fisheries of the region registered a total weight of blue marlin landings of 1376.2 kg between 2017 and 2019, resulting from commercial fishery bycatches.

It was also possible to compare the average weights of specimens captured in each of the Macaronesian archipelagos and the numbers of individuals caught, owing to the registers at billfishreport.com (accessed on 20 May 2020). The results indicated similar average weights in all the studied regions (Table 5) where large specimens were usually captured, but in terms of individuals captured, the numbers for Cape Verde were higher than for any other Macaronesian archipelago.

### 3.5. Relationship between CPUE of Blue Marlin and Environmental Variables

The relationship between the explanatory environmental variables and the CPUE of blue marlin (response variable) is shown in Table 6. We first fitted unadjusted models onto the available predictors and then obtained an adjusted model (i.e., saturated) via the forward stepwise procedure. Since the results of both adjusted and unadjusted models

were similar, except for the absence of the NAO in the adjusted model, the unadjusted models are shown in Figure 5.

**Table 5.** Weights in kg (average, SD—standard deviation, *n*—number of individuals, and number of boats) of blue marlin in the Macaronesian region from 2011 to 2019 (data collected from billfishreport.com, accessed on 20 May 2020).

| Region | Average Weight (kg) | SD | *n* | Number of Boats |
| --- | --- | --- | --- | --- |
| Madeira | 307.79 | 58.57 | 419 | 31 |
| Cape Verde | 290.13 | 60.24 | 825 | 29 |
| Azores | 300.52 | 62.05 | 365 | 17 |
| Canary Islands | 276.94 | 51.77 | 350 | 38 |

In addition to the strong monthly variation in blue marlin CPUE, which showed the strong seasonal character of this fish, both SST and cloud cover influenced blue marlin catches. While the cloud cover negatively influenced the catches (higher CPUE values were obtained with sky coverage of ≤50%; Figure 5), SST was positively correlated with catches, reaching an optimum between 22 and 23 °C. Moreover, the NAO may have influenced the catches of this species, since a significant relation was observed in the unadjusted model, wherein elevated NAO values seem to have adversely affected CPUE (Figure 5).

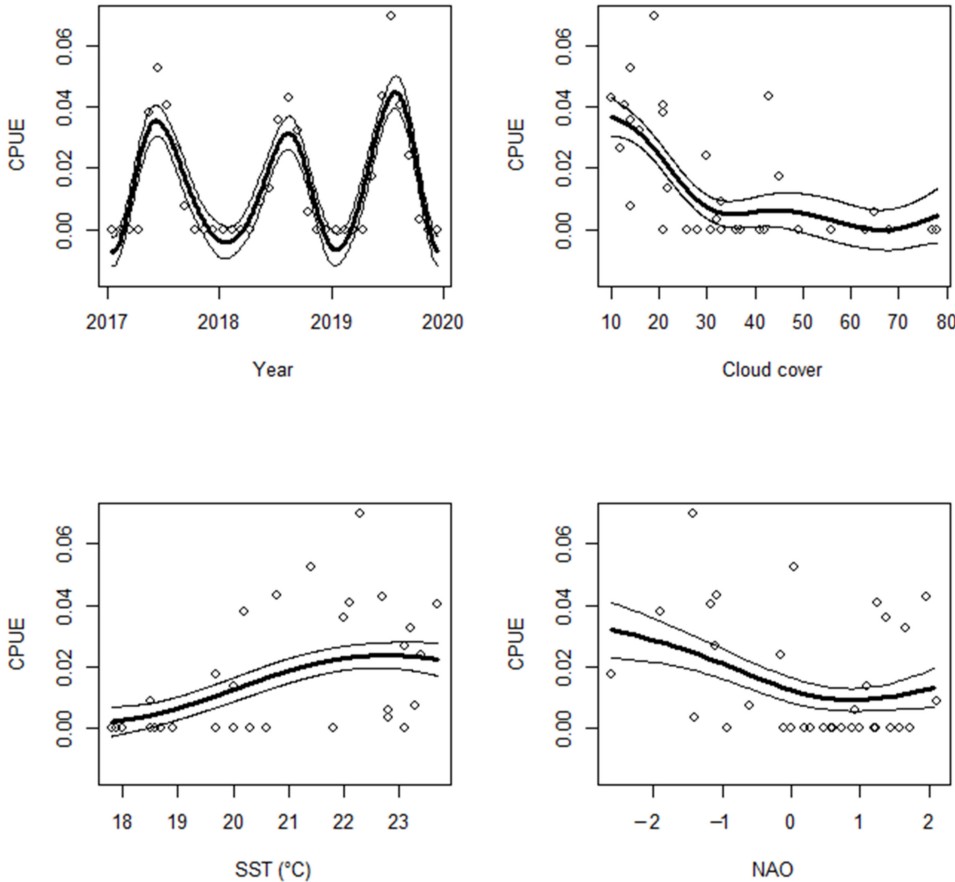

**Figure 5.** Partial effect of considered factors (year, cloud cover, SST, and NAO) on blue marlin captures. Number of blue marlins caught (points), the prediction (dark lines), and their standard deviation (thin lines) estimated by the GAM unadjusted model.

**Table 6.** Response and explanatory variables, explained deviance, and Akaike's information criterion (AIC) of the five generalized additive models (GAM) fitted on the CPUE of blue marlin between 2017 and 2019. Significant unadjusted models (top four) and the final adjusted model (bottom, that included three variables plus interaction terms), all of which were fitted with a Gaussian error distribution, are shown.

| Response | Explanatory | *p*-Value | Deviance Explained | AIC |
|---|---|---|---|---|
| CPUE | Month | <0.0001 | 78.00 | −218 |
| CPUE | Cloud cover | 0.0002 | 47.40 | −195 |
| CPUE | SST | 0.0007 | 30.90 | −189 |
| CPUE | NAO | 0.0221 | 18.70 | −183 |
| CPUE | Month | 0.0008 | 92.50 | −246 |
| | Cloud cover | <0.0001 | | |
| | SST | 0.0002 | | |
| | Month*Cloud cover | 0.0074 | | |
| | Month*SST | 0.0036 | | |

## 4. Discussion

Blue marlin captures registered from 2008 to 2019 showed considerable variability between years, with a high number of individuals being caught in the early years of the period, followed by a strong decrease and far fewer individuals being caught. Finally, in the last year, the numbers almost returned to those of the first year. Unfortunately, it was not possible to obtain the records of fishing trips and the numbers of boats involved for the first few years, and it was therefore not possible to make accurate comparisons. It was only possible to calculate the monthly CPUE from 2017 to 2019.

The obtained CPUE results seem to agree with the basic features of recreational fishing: high fishing effort and low catch rates [27]. As observed, the probability of catching a blue marlin is quite variable. This is supported by the differences in CPUE obtained for the different boats and their variations from year to year, including the high CPUE value of 0.062 obtained in 2008 for the region by Graça [13]. This higher value observed in 2008 corresponded to a high number of captures of blue marlin; however, the higher CPUE could also be related to the low number of boats (five) analyzed in that study.

Some previous studies of the catch rates and effort of recreational fisheries also registered a very variable CPUE in other areas of the Atlantic Ocean, such as in the Azores [28], the Maryland coast [29], and in the Gulf of Mexico [30–33]. These works confirm the high interannual variability in CPUE, which is probably related to the migratory behavior of this species, as blue marlin travel long distances in search of warmer waters and follow the currents, which can vary from year to year.

According to the latitude of each archipelago, the variability in seasonality allows us to infer the probable migratory route of this species in this geographical area. The results from the analysis of catches around different Macaronesian archipelagos suggest the presence of blue marlin in equatorial waters during the early winter and the northward migration in late winter and early spring, reaching Azorean waters in the summer. Therefore, we can assume that as the water temperature rises over the course of the year, this species begins its migration to northern latitudes, probably for reproductive purposes. Our results are consistent with those of previous studies in which similar blue marlin migration patterns were observed in other parts of the world, such as in the Pacific (indicating that this species migrated northward during April–October, and south thereafter [8]) and in the southern Atlantic region, where blue marlins were present in the south during the austral summer time and then moved north toward warmer equatorial latitudes during the winter [34–36].

Recently, several studies on this species have been carried out using pop-up satellite tags [1,12,34,36,37], which provide data on movements, distribution, and post-release survival in different areas of the world. The results for this geographic area are currently being analyzed (Freitas et al., unpublished results), and this will allow us to confirm whether the migration movements suggested in this work are correct.

Analysis of blue marlin seasonality in Madeira showed that they always arrive at the same time of the year (end of May, beginning of June), when the waters become warm (19–20 °C). The SST where blue marlins were caught ranged from 18.5 to 24.5 °C. This is similar to the range observed along the south coast of Portugal, where captures were registered between 18.6 and 25.5 °C [38], but lower than that registered by Crespo [36] for the Southwest Atlantic. In that region, an SST range between 24 and 29 °C was observed for 90–92% of the cases, and blue marlins moved southward off the Brazilian coast in order to spawn following the displacement of the 25 °C isotherm in the summer [35].

The results from Madeira reflect that in June, when the catches were most abundant, the average temperature was 20.7 °C, while in September the water reached the highest temperature (with an average of 23.7 °C) but blue marlins were sparse. These data indicate that blue marlins seem to prefer warm waters, but it is not necessarily the case that there is a greater abundance of specimens when the water is warmer. Therefore, this association is probably due to the greater abundance of this species in the summer months, which is when the water is warmer, and thus the probability of catching a specimen is higher.

The SST appears to be positively linked with catch rates of pelagic species in sport fishing in the Pacific Ocean [39], but it has a relatively minor influence on the CPUE of the Brazilian longline fleet [40], a detail that could be masked because marlin was considered a bycatch species in that study; direct fishing might yield different results. Additionally, Carlisle et al. [1] suggested that the horizontal distribution of blue marlin in the central Pacific was influenced by SST and large-scale fluctuations thereof, in particular those associated with strong La Niña conditions, which might influence marlin migratory behavior. The eastern Pacific Ocean's blue marlin population moves east during El Niño years, as evidenced by catch rates [8]. This behavior is supported by studies [41,42] indicating that the distribution and movement patterns of tuna-like species may be strongly linked with environmental variation, such as El Niño–Southern Oscillation (ENSO) events and related changes in various oceanographic features. However, the present study was carried out in the Northern Hemisphere; therefore, the relation between blue marlin presence and NAO was evaluated instead, as it seemed likely that it would influence blue marlin catches.

NAO is considered the largest source of variability in climate oscillation affecting the North Atlantic region, redistributing air mass from the Arctic to the subtropical Atlantic [43]. The variability introduced by the NAO affects the ocean by changing many parameters, varying the SST, the depth of the mixed ocean layer, the ocean heat content, sea ice cover, surface current circulation, the intensity and direction of the prevailing winds, and several meteorological phenomena such as rain and storms [44,45]. During positive NAO periods, the conditions are cooler and drier than average in the Northwest Atlantic and Mediterranean regions, while conditions in northern Europe, the eastern United States, and parts of Scandinavia become warmer and more humid than average [44]. Some factors influenced by the NAO increase the presence of nutrients in the sea, thus altering the trophic levels of marine ecosystems and their exploitable resources [45,46]. This could affect the catch of large migratory pelagics, as westerly winds originating from a positive NAO could displace the schools eastward towards the European and African coasts, as suggested by Rubio et al. for two species of tuna [45].

Additionally, higher catches were associated with low cloud cover. This result should be considered with caution, since we have to consider that the fishing season is concentrated in the summer months, and therefore will include few days with cloudy skies. In addition, fewer fishing trips were probably taken on such days due to the possibility of rain. A higher CPUE is associated with the presence of less cloudiness; therefore, the catchability increases with the level of light in the water. This may be related to the fact that blue marlins are visual predators [47], and it is easier for them to see the lure on clear days. This is exploited by fishers to increase their chances of catching blue marlins, as they adapt their fishing equipment according to the weather conditions. On very calm days, the speed of trolling is higher and the line used is thinner in order to make it invisible, while in bad

weather with higher waves the line tends to jump instead of sinking, so a thicker line with larger and heavier lures is used in order to make it sink more efficiently and be more visible to marlins (fisher's comment).

Finally, it is interesting to note the possible effects of fishing practices on blue marlin populations. Because of the practice of C&R carried out by the big game fishing fleets, we could assume that this fishing practice might not have a significant negative impact on the target species. Until recently, there was an almost complete lack of knowledge on the effects of C&R on the survival of most fish species, but some studies have observed that the mortality of blue marlin subjected to C&R is low (89% survival after tagging [12]), which suggests that C&R is a viable management option that protects populations [48–50]. Nevertheless, it is still a practice that raises some concerns, as handling can cause great stress and lead to subsequent death for fish caught and then released. Various factors, such as hooking on internal organs, the removal of hooks from deeply hooked fish, the depth at which fish are caught, water temperature, and handling time, can all contribute significantly to mortality [14]. In order to assess the real impact of this practice, it would be useful to carry out a study on the delayed mortality of blue marlin following a normal C&R protocol.

Some studies have shown that the most serious impact on blue marlin is made by longlines used for tuna and swordfish [4,29,51], suggesting that longline fishing should be restricted in seasons and areas with high blue marlin CPUE [52] so as to reduce the fishing mortality of this species. However, this management measure is difficult to apply in Madeira, since these fishery practices usually coincide in both season and fishing area. The immediate release of the fish after capture would help reduce mortality and increase the resilience of the populations [29]. Pelagic longlines in Madeira are not very common; a pole and line with live bait is the most frequently used tuna fishing method in the region, while pelagic longlines directed at black scabbardfish are also employed, but these are placed deep in the water, and only deep-water species tend to be captured as bycatch or incidental species [53,54]. Therefore, these methods do not pose a significant risk to species such as marlin. This was confirmed by the infrequent landings of blue marlin registered as bycatch in the regional commercial fisheries, where only the equivalent of 1376.2 kg was landed between 2017 and 2019 (Regional Directorate of Fisheries), considerably less than the almost 7 tons (4.2% of the total catch) estimated for the recreational fishery in the same period. These results point to the relatively low mortality of blue marlin in the region, suggesting that regional fisheries may have an overall low impact on this species.

Average weight of blue marlin captured between 2010 and 2019, 290.5 kg ($n = 938$), was very close to the value obtained by Graça [13] for the same region (298.7 kg), and slightly higher than the 277 kg registered for southern Portugal [38]. If we compare this with the average weights of blue marlin from other parts of the world, it is interesting to observe that larger specimens were caught in Madeira than in either the western Atlantic (with an average of 236.6 kg off the Maryland coast [29] and 177 kg in the northern Gulf of Mexico [30]) or the Pacific (with an average weight of 175 kg [2] and 108.7 kg in Baja California [32] or 155 kg in New South Wales [55]).

For the eastern Atlantic, there is little information about this fishery in the Azores [28] beyond the average weights of blue marlin landed in 1985 (157.3 kg) and 1986 (210 kg). The analysis performed for the Macaronesian region showed that the average weight in Madeira is similar to that in the other archipelagos, suggesting that the individuals captured could be from the same population.

The other Istiophoridae species captured was the white marlin, but fewer catches of this species were registered. This species normally arrives in Madeiran waters a little earlier than blue marlin, usually being captured between May and November, but there was no significant presence as bycatch until June to August period. Some other big pelagic fishes were captured included dolphinfish (with a peak in catches during July and August), some wahoos (mainly during the second half of the year, occurring more frequently from

August to October), and different tuna species, the most frequently caught of which was bigeye tuna, which were found mainly from April to September.

Some of these other species captured by the big game fishing fleet (tuna, dolphinfish and wahoos) are usually landed, since many of them are highly valued as food. The results show that a relatively small number of fish were caught, but it would be interesting to study these catches and landed specimens. Only occasionally, as in 2018 with dolphinfish, were large numbers of individuals caught during the same fishing trip, due to small concentrations of this species being found below floating objects and following ships [56]. It would be interesting to conduct a more detailed follow-up study including the systematic sampling and weighing of these other species in order to better understand certain characteristics that would help to assess the impact of this fishery practice on the target species.

Finally, it is important to highlight the problems associated with data acquisition in the present study, in light of the importance of reliable and consistent data collection for adequate analysis. Unfortunately, the authors were faced with many difficulties when compiling the information for this study. The initial aim was to register fishing trips and catches using data sheets, as this is considered one of the easiest and least expensive sampling methods [57]. The problem is that their completion is perceived as demanding by fishers [58], and as a result, there was little acceptance on the part of captains.

Additionally, some crews were unwilling to cooperate, and only agreed to participate after some insistence. It was necessary to establish periodic contact to share the data and build confidence over time, which facilitated the collection of more accurate information. Furthermore, it was difficult to obtain historical data for this fishery due to the absence of logbooks, and only catch registers recorded by one of the authors for personal use were available.

The data obtained in this study should therefore be used only as a very rough indicator of fish population size. In addition, abundance indices obtained from catches (such as CPUE) are difficult to interpret due to the limited understanding of how changes in fishing strategy interact with the behavior and distribution of marlin within the water column depending on the area and season [59]. Furthermore, the absolute number of recorded fish may be affected by the different amount of effort expended in fishing, which will correlate with the number of boats and the hours trolled and which may vary from year to year.

In conclusion, this is the first study of the practice of big game fishing in Madeira, and it contributes to expanding our knowledge about the seasonal distribution of blue marlin. The presence of blue marlin in Madeiran waters is markedly seasonal, and is probably related to their migratory behavior and some of the environmental factors analyzed. Nevertheless, no determining factor has been found that can satisfactorily explain the greater catches of blue marlin in certain years. Therefore, more studies should be carried out, and a program for monitoring this activity should be implemented. This work shows the difficulties encountered in obtaining data from this fishery and the importance of consistently gathering data on catches over time. People practicing this fishing modality should be involved in these studies as stakeholders, collaborating with authorities and researchers in order to obtain accurate data. This would help in maintaining adequate records and monitoring how the fishery practice and target species evolve over time.

**Author Contributions:** Conceptualization, R.M.-E. and M.L.F.d.G.; Data curation, R.M.-E. and E.T.; Formal analysis, R.M.-E. and P.P.; Funding acquisition, M.L.F.d.G. and N.M.A.G.; Investigation, R.M.-E.; Method-ology, R.M.-E.; Project administration, R.M.-E.; Resources, M.L.F.d.G., N.M.A.G. and E.T.; Super-vision, M.L.F.d.G., N.M.A.G. and M.H.; Validation, P.P. and M.H.; Writing—original draft, R.M.-E.; Writing—review & editing, R.M.-E., P.P., M.d.F. and M.H. All authors whose names appear on the submission made substantial contributions to the conception or design of the work; the acquisition, analysis, and interpretation of data; and the creation of new software used in the work. All authors have read and agreed to the published version of the manuscript.

**Funding:** Roi Martínez-Escauriaza was financially supported by the Oceanic Observatory of Madeira Project (M1420-01-0145-FEDER-000001-Observatório Oceânico da Madeira-OOM). Margarida Her-

mida was financially supported by a postdoctoral grant from the Regional Agency for Development of Research, Technology and Innovation of Madeira (ARDITI), project M1420-09-5369-FSE-000001. Pablo Pita was funded by the *Xunta de Galicia* (RECREGES II project under Grant 1400 ED481B2018/017), by the H2020–European Comission project *Atlantic ECOsystems assessment, forecasting & sustainability* (AtlantECO, ref.2019-PI022), and *EQUALSEA* project ERC Consolidator, under Grant Agreement nº101002784 funded by the European Research Council. PP also acknowledges economic support of the *Xunta de Galicia (Grupo de Referencia Competitiva* GI-2060 AEMI, under Grant 1401 ED431C2019/11), and of the project *Grupo de Trabajo sobre Pesca Marítima Recreativa en España,* funded by the Fundación Biodiversidad of the Spanish Ministerio para la Transición Ecológica y el Reto Demográfico, co-funded by the European Maritime and Fisheries Fund.

**Data Availability Statement:** Madeira sea surface temperature (SST): https://www.esrl.noaa.gov/psd/ (accessed on 13 September 2020); Madeira historical weather conditions: https://power.larc.nasa.gov/data-access-viewer/ (accessed on 13 September 2020); North Atlantic Oscillation (NAO): https://www.ncdc.noaa.gov/teleconnections/nao/ (accessed on 13 September 2020).

**Acknowledgments:** This work could not have been completed without the collaboration of the crews and people who selflessly offered data on this fishery and catches in recent years. Thanks for all the information offered, the help received in better understand this fishing modality, and the chance to come onboard in pursuit of these large animals. Thanks to Fernando Alexandre for the native English language revision, and to Francesca Gizzi for helping improve the manuscript and map elaboration. Additional thanks go to the anonymous reviewers who provided comments on an earlier draft of the manuscript.

**Conflicts of Interest:** The authors declare no conflict of interest.

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
