# Peer review of "Analysis of Big Game Fishing Catches of Blue Marlin (Makaira nigricans) in the Madeira Archipelago (Eastern Atlantic) and Factors that Affect Its Presence"

_sustainability, doi:10.3390/su13168975_

Round 1

Reviewer 1 Report

As the authors claim in the conluding paragraph "this work is the first study of the practice of big game fishing in Madeira". It sounds interesting in theory, but the actual study has lots of flaws, in my opinion. 

My strongest objection is the quality of the collected data. There are many "assumptions" and "estimations" and differnet ways of collecting data, which makes the analysis rather inaccurate and unreliable.

My suggestion to the authors would be to just focus on describing the main characteristics of the "Bue Marlin big game fishing in Madeira" and do exactly that! No need to complicate things by adding "environmnetal factors". Describe the fleet, fishing patterns and techniques, how many people are actually on board fishing (not mentioned anywhere), total catch, lures used (just to mention a few).

Also, the use of English language really needs improving.

For more detailed comments and remarks, see the comments within the attached .PDF file

Reviewer 2 Report

This manuscript is a well written manuscript that reports an analysis of the big game fishing catches of blue marlin at Madeira archipelago, Eastern Atlantic and discuss some factors that condition its presence. The data collection is appropriate and the analysis is valid. The discussion is well elaborated and supported by references. I just have two minor comments.

  1. One of them is related to the current status of the blue marlin population in the East Atlantic. The authors argues that the 4.2% landing of blue marlin have an overall low impact on the population without mentioning the size of the (or current status) of the population.
  2. My second comment refers to figure 2 and whether "lost fish" is fish that was hooked by not landet.

Reviewer 3 Report

Overall

The grammar and writing style of the manuscript needs significant editing. Some sections of the manuscript were almost impossible to interpret because of poor grammar.

This study is preliminary at best. Much of the data presented prior to 2017 is unreliable and should not be used to model with environmental conditions. There are other analyses that could have been explored to evaluate some of the study limitations. I recommend conducting some of these additional analyses so that readers can have more confidence in your results.

Introduction

First paragraph of the introduction: You say that they are generally found >24C, but then provide evidence of >22C from the Atlantic. How are these two facts linked together?

I don’t think the second paragraph (and much of the third paragraph) in the introduction is needed. This is just a simplistic description of big game fishing, which doesn’t really add anything to the manuscript. I think it would be better to focus on catch statistics and fisheries regulations, both worldwide and in your study region.

The introduction needs significant revision, both for content and structure. I’d suggest expanding on what is known of blue marlin biology, ecology and migration to provide context for your study. Also, it would be worthwhile defining your study site more, particularly has it related to oceanographic conditions (as these are important to pelagic fishes). Given what is known about blue marlin elsewhere, what are the true knowledge gaps in your region?

How does objective (b) differ from objective (a)? Also, for objective (c) and (d), you should provide background knowledge of these topics in the introduction to help set-up the paper.

Methods

Do you have specific locations of fishing effort and blue marlin catch? If so, you might consider presenting it as a map similar to figure 1.

From 2009-2017, were trips where no fish were caught recorded? Or was it only for trips 2017-2019?

For your regression analysis (2010-2019), did you use total catch or presence data? Also, given that these change with fishing effort, how closely do you think catch aligns with abundance?

Given that you had CPUE and total catch for some years, could have you compared the two data types to see how strongly they were correlated? This may help inform your analysis using earlier data without CPUE.

For GAM, is daily occurrence of blue marlin an appropriate time scale for the explanatory variables? I would imagine that daily values are a too-fine of temporal scale for the type of data you have.

Results

The first few sections of your results focus in on the time-frame 2017-2019, when you have more complete data. If this is the focus of your study, could you just exclude data from before 2017? If not, you might consider introducing data/results from these earlier years in the first part of your results.

You say that blue marlin was the species with the highest catches historically in the fishery. Is this correct? I would imagine it would have one of the lowest catch-rates. Can you present data to support this?

Figure 3 shows quite a bit of inter-annual variability in total blue marlin caught. I wonder what drives this. Also, I’m not sure it is appropriate to use the word “recovery” when describing this as it may not be linked to overfishing or anything like that.

Is the weight analysis based on actual weights of dead animals, or estimated weight of released fish (or both)? If weights are based on fish killed during tournaments, these may not be representative of the population.

For Table 4, can you reduce this list to just a few key models? The remaining models could be presented as supplementary information.

I would imagine that the environmental parameters are fairly reliable. In contrast, it is likely that blue marlin catch numbers and the catch of prey species don’t accurately represent the abundance of these species. As such, I’m not really sure how much information we can draw from these analyses.

Discussion

You might consider restructuring the discussion to start with an overview of your key findings, rather than your limitations.

The limitations of the data and methods are significant and I don’t think they have been well investigated or adjusted for. As such, I don’t think these analyses present any reliable insights into the blue marlin fishery

There were some limitations and exceptions described in the discussion that could have been explored in the data. For example, the authors could have conducted some modeling with fishing effort as the response variable to see how things like weather influenced the number of fishing trips.

Generally, the scope of the discussion is OK. The discussion is quite long given the reliability of the results. The authors might consider making the discussion more concise and appropriate for the preliminary nature of this study. This section, like the entire manuscript, needs significant revision of grammar and writing style.

Round 2

Reviewer 1 Report

The authors really improved the manuscript.

Still, there are a few matters which require revision. See comments in the attached pdf.

Author Response

-According to the authors' reply they stated that they modified the title, but the word "condition" is still there. I would (again) suggest the use of a better word, such as "affect"

This is the title from the previous version of the manuscript. The authors should make sure they change the title here, too (preferably with a different word than "condition")

Agreed. We changed “condition” to “affect” (in both places).

-I don't know if this a formating problem from the template or the authors missed a space between Genus and Species.

It was indeed a formatting problem, and we checked all the text.

- Abbreviations should be written in full when first mentioned and then used freely in the rest of the text. Thus, the authors should explain what CPUE means and then use the abbreviation.

Agreed. We followed the reviewer’ suggestions.

-"Blue marlin" and "catch" are in singular form, so it is more appropriate to use "is" here

Agreed. We changed the text accordingly.

- Add the abbreviation (C&R) since the authors are using it in the next sentence.

Added in the text.

-again singular form, so "making IT one of the..."

Agreed. We changed the text accordingly.

-In the previous version of the manuscript the map was not a useful addition to the text, so I didn't mention that it needs a mini-map to help the readers see where in the world is the Study Area.

Since the authors changed the caption and the reference to the map in their text, they should add a small mini-map in the top-left corner with a bigger part of the Atlantic Region (preferably with parts of the European and African coasts visible) and an annotation (square, circle, arrow or something else) to pinpoint the study area within the region (for example, the map in Figure 4)

We modified the figure following the reviewer’ suggestion.

- space needed between words (could a formating error but the authors should double-check to make sure)

We added the space.

- "collected" is more appropriate. The authors gathered the registered results from their sources.

Changed in the text.

- from 'the webpage" billfishreport.com or from "the online database" bllfishreport.com

Agreed. We changed it to “the online database billfishreport.com”.

- this looks like discussing the results from the analysis of the collected data. I think it should be moved to the discussion section or rephrased to be more appropriate for the Materials section

Agreed. We restructured this phrase and included part of this information in the discussion section.

- suppose this is "milli bar"? I think the official unit (at least in Europe) for Atmospheric pressure is Pa (pascal) so maybe the authors should use the hecto-pascal (hPa) unit which is the equivalent of the milli bar.

We changed the unit in the text.

- Should go after "North Atlantic Oscillation Index"

Agreed. We changed it in the text.

- no need for quotes

Removed.

- space between

Corrected in the manuscript.

- My comment in the previous version of the manuscript was

"- A table or a graph would help the readers to visually understand the size of the fleet and its

fluctuation over the years. Also the different attributes (such as Private or Charter, LOA, Home

Marina) would make more sense."

And the authors' reply was

"Agreed. We introduced a Table describing the fleet."

This table is even more complex to understand and it is not actually "describing the fleet". This looks like the raw data which the authors collected and it is providing any useful info. For example, what would be the meaning in comparing (let's say) Boat 1 with Boat 14??? The first did 40 trips in 2017 and the other 42...So what? Are there any significant qualitative differences between the boats that would give meaning to that comparison (for example, charter vs private, Marina 1 vs Marina 2)

My suggestion to the authors would be to only present the sum of their data, but in a more constructive way: Total trips (and other variables) for Private and Charter boats from Marina 1 and for Marina 2, projected in the span of the 3 years. That would be a simpler table (or graph), more easy to comprehend and more meaningful. And also, it would visually explain that there are no significant differences between charter and private boats, thus justifying the use of the pooled data in the other analyses.

We modified the table using the data more constructively, as suggested by the reviewer. In order to have a good data comparison between the two marinas and the two boat categories (Private and Charter), we decided to show the average and not the sum due to the difference in boat number. However, we added the total sum of the data at the end of the table.

- Table captions should be place at the top of the Table

We changed the position of the caption.

- If the authors chose to keep this table (which I don't suggest) they should explain all the column titles/abbreviations. For example, what does "No Fishes" mean? Total number of individuals or number of different species of fish caught?.

We decided to follow the reviewer’ suggestion and deleted the table, while adding the new one suggested above.

- By reporting the sum of their data, the authors justify my comment regarding the complexity of Table 2.

Agreed. We modified the table as suggested before.

- Blue Marlin individuals (or "specimens" which I don't prefer but the authors seem to use it frequently)

We added “individuals” in the figure caption.

- Discussion of the results. The fact that the authors chose to close the paragraph with the word "conclusively" makes it very obvious! I suggest moving the paragraph to the Discussion

Agreed. We moved part of the paragraph to the Discussion.

- space between

Space added.

- Remove "a".   average weight of Blue Marlin

Corrected in the manuscript.

- What is the purpose of this paragraph? No results are being presented! It sounds like some more details on what happens in Big Game Fishing...

Why do the authors mention what happens to the stomach and tissue samples? Did they perform or present the relevant analyses?

I should have made that comment in the previous version, but somehow it escaped my attention.

Agreed. We deleted the paragraph.

- no need to cite Figure 3.

Agreed. We removed the citation.

- data collected from billfishreport.com

We changed the text accordingly. Please see above.

- space between

Corrected.

- which factors?

The authors used 4 specific factors, they should present them in the caption

Agreed. We changed the caption accordingly.

- “Other species” last paragraph: Discussion

Agreed. We moved this information to the discussion section.

- no need to cite Results or figures in the Discussion

We removed the citations, as suggested.

- either "Blue Marlins ... arrive"

or "Blue Marlin .. arrives"

We changed the text accordingly.

- The authors keep interchanging between marlin and blue marlin. It is difficult to understand if they refer to all Marlin species or just Blue Marlin... they should try and be more consistent to avoid misunderstandings of their results/conclusions

Agreed. We specified the species in the entire manuscript to avoid misunderstandings.

- Again, does this refer to Marlins in general of specifically to Blue marlins???

This is now specified in the text.

- Could be omitted.

We deleted it.

- Space between

Corrected.

- "the authors"

The authors are referring to themselves, not to researchers in general

Agreed. We changed the text accordingly.

Reviewer 3 Report

The authors have done a good job of incorporating my comments and suggestions into their manuscript revisions. The writing and grammar are much better, but it still needs some minor editing in the proof stage. I also encourage the authors to check all figure and table captions to make sure they are adequately descriptive. I have provided some more specific comments below:

Introduction

The authors have done a good job at adding detail to their introduction. However, some sections of the introduction need to be re-arranged so that it flows better. For example, the first part of the introduction that discusses water temperature isotherms could be moved further down in the introduction when they talk about distribution, etc.

The revised text in the introduction discusses fisheries and catches, yet the authors do not provide any statistics. I suggest that they provide some global statists from FAO or similar.

The second paragraph of the introduction is repetitive and vague, and doesn’t’ actually provide much information on what has been published on Blue Marlin in the region. It should be revised and moved, or deleted.

The fourth paragraph, that mentions other pelagic species, is not needed.

Paragraph 5 is good, but I’d recommend including more specifics about how Blue Marlin are targeted and captured in the fishery, including statistics on catch and effort.

With regards to the objectives, what were catch data analyzed for? Spatiotemporal trends? Population dynamics?

Methods

Much of the information provided in paragraphs 1, 3, 4 and 5 of the methods seems repetitive. I’d suggest the authors try to make a description of the fishery data more concise.

I think the authors have done a good job revising their analyses so that good-quality data are used for the correlations, and that a monthly time scale is used for the correlations.

When trying to correlate fish presence or behavior with environmental parameters, there can often be a lag effect (e.g. rainfall a month or two before may have driven production in the area and lead to increased fish abundance a month or two later, rather than in the same month). Did the authors test this?

I see that the authors used additional data for part of their paper that was not included in the regression (e.g. website data for length-weights). The authors may consider moving this data description to earlier in the methods (because it is more preliminary). Also, if there are other data that the authors used for observation and inference, but were not part of the GAMs, then they may include them here too.

Results

For Figures 2-4, total catches can be misleading when they are not adjusted for effort (i.e. CPUE). Can the authors re-plot these graphs using CPUE rather than total catch?

For Figure 2, I wonder if the authors can overlay mean monthly SST to display what drives this seasonality.

For Figure 3, do the authors discuss why there was this reduction, and subsequent increase, in catch rates?

In table 3, switch the columns “Min” and “Max” so that Min is before Max.

Why are there 886 shown in table 3, but then only 419 shown in table 4? Is this because they come from 2 separate datasets?

Table 5 is a little difficult to interpret. How many models are shown here? I’m thinking 5 models, the first 4 models are single-factor models and the final model has 3 explanatory variables – is that correct? Is there a clearer way to show this? Also, when showing the results of multiple models, typically we present the best-fit model first, then other close models after, with a delta-AIC (rather than actual AIC).

Did the authors try and models with interaction terms (e.g. month & SST). I imagine there are strong interaction terms with some of the explanatory variables.

Section 3.7 might be better placed elsewhere in the results, or even as supplementary information. It doesn’t add much to the manuscript.

Discussion

The authors have made good revisions to the discussion. They may also consider adding subheadings to the discussion to help organize the sections more clearly.

One comment I had on the previous manuscript was for others to consider exploring other response variables with their models (e.g. fishing effort). This might provide insight into how weather conditions influence fishing activity. The authors say that they addressed this comment by including fishing effort in their analysis, but I don’t see this reported in the results. Perhaps I’m missing something.

Author Response

The authors have done a good job of incorporating my comments and suggestions into their manuscript revisions. The writing and grammar are much better, but it still needs some minor editing in the proof stage. I also encourage the authors to check all figure and table captions to make sure they are adequately descriptive. I have provided some more specific comments below:

Introduction

- The authors have done a good job at adding detail to their introduction. However, some sections of the introduction need to be re-arranged so that it flows better. For example, the first part of the introduction that discusses water temperature isotherms could be moved further down in the introduction when they talk about distribution, etc.

 Agreed. We moved this part of the introduction as suggested by the reviewer.

- The revised text in the introduction discusses fisheries and catches, yet the authors do not provide any statistics. I suggest that they provide some global statists from FAO or similar.

Agreed. We addedthe few available information from the FAO, including the state of blue marlin populations and some related statistics.

- The second paragraph of the introduction is repetitive and vague, and doesn’t’ actually provide much information on what has been published on Blue Marlin in the region. It should be revised and moved, or deleted.

 We deleted the paragraph accordingly.

- The fourth paragraph, that mentions other pelagic species, is not needed.

Although we understand the suggestion of the reviewer, in this instance we decided to maintain this paragraph because itintroduces 4 species that are usually caught together with the blue marlin. Moreover, the 4 speciesare mentioned again in the results and discussion sections.

- Paragraph 5 is good, but I’d recommend including more specifics about how Blue Marlin are targeted and captured in the fishery, including statistics on catch and effort.

We thank the reviewer for the suggestion, but we deleted this information from the first version following a suggestion from another reviewer. However, we added a sentence describing the procedure, and we included the FAO statistics available for the Atlantic region.

- With regards to the objectives, what were catch data analyzed for? Spatiotemporal trends? Population dynamics?

Agreed. We specified the objectives more clearly.

Methods

- Much of the information provided in paragraphs 1, 3, 4 and 5 of the methods seems repetitive. I’d suggest the authors try to make a description of the fishery data more concise.

Agreed. We restructured this section accordingly. 

- I think the authors have done a good job revising their analyses so that good-quality data are used for the correlations, and that a monthly time scale is used for the correlations.

We thank the reviewer for the comment.

- When trying to correlate fish presence or behavior with environmental parameters, there can often be a lag effect (e.g. rainfall a month or two before may have driven production in the area and lead to increased fish abundance a month or two later, rather than in the same month). Did the authors test this?

We agreed that it would be interesting to test this. However, due to the weakness of the presence records, we decided to focus only on the specimens captured. 

- I see that the authors used additional data for part of their paper that was not included in the regression (e.g. website data for length-weights). The authors may consider moving this data description to earlier in the methods (because it is more preliminary). Also, if there are other data that the authors used for observation and inference, but were not part of the GAMs, then they may include them here too.

 We restructured this section following the reviewer’ suggestions. 

Results

- For Figures 2-4, total catches can be misleading when they are not adjusted for effort (i.e. CPUE). Can the authors re-plot these graphs using CPUE rather than total catch?

We agree with the reviewer that it would be more correct to use CPUE instead of total catches; however, we only have CPUE data for three years (2017-2019), and consequently, we would lose information from the previous years. For this reason, we have decided to maintain the total catches.

- For Figure 2, I wonder if the authors can overlay mean monthly SST to display what drives this seasonality.

We have replaced Figure 2 with a new version including the mean monthly SST.

- For Figure 3, do the authors discuss why there was this reduction, and subsequent increase, in catch rates?

We thank the reviewer for the comment. However, we don´t have enough information to discuss the reason for these fluctuations. We mentioned in the discussion section that the fluctuation of the catch rate could be related to the presence of these animals in the region with the current that they follow, which can change from one year to the other, but we did not feel confident speculating on other possible causes.

- In table 3, switch the columns “Min” and “Max” so that Min is before Max.

 Agreed. We changed it accordingly.

- Why are there 886 shown in table 3, but then only 419 shown in table 4? Is this because they come from 2 separate datasets?

In table 4 we only use the data collected from billfishreport.com and in table 3 are included registers from all the sources.

- Table 5 is a little difficult to interpret. How many models are shown here? I’m thinking 5 models, the first 4 models are single-factor models and the final model has 3 explanatory variables – is that correct? Is there a clearer way to show this? Also, when showing the results of multiple models, typically we present the best-fit model first, then other close models after, with a delta-AIC (rather than actual AIC).

The reviewer is right. We are showing 5 models in Table 6 (Table 5 in the previous version), the first 4 are single-factor models (unadjusted), and the last one with 3 explanatory variables (adjusted). This representation is quite standard, and we don’t see how could be easily improved. However, we have included some changes in the caption that we hope will help readers to understand this.

Regarding the order of the models in the table, as we explained in the text (section 3.5. Relationship between CPUE of blue marlin and environmental variables) we first obtained unadjusted models, and by a forward stepwise selection procedure we obtained the adjusted model (saturated). This is why we followed this order in Table 6.

Finally, we prefer not to show differences between the AIC (delta) because from the true value of AIC the readers are able to make their own comparisons. All the models have been obtained by the same algorithms and are nested, so can be safely compared.

- Did the authors try and models with interaction terms (e.g. month & SST). I imagine there are strong interaction terms with some of the explanatory variables.

Following the reviewer’s suggestion, we now include two interaction terms in the adjusted model. See Table 6. 

- Section 3.7 might be better placed elsewhere in the results, or even as supplementary information. It doesn’t add much to the manuscript.

Agreed. We moved the section 3.7 after the CPUE section. We decided to insert here this section because, in the CPUE section, we distinguish the total CPUE where all species were used for the calculation and the CPUE of blue marlin.

Discussion

- The authors have made good revisions to the discussion. They may also consider adding subheadings to the discussion to help organize the sections more clearly.

We thank the reviewer for the comment. We tried to follow the suggestion, but in the end, we decided to leave the discussion without sections to maintain more fluent text that sometimes is interconnected between different sections.

- One comment I had on the previous manuscript was for others to consider exploring other response variables with their models (e.g. fishing effort). This might provide insight into how weather conditions influence fishing activity. The authors say that they addressed this comment by including fishing effort in their analysis, but I don’t see this reported in the results. Perhaps I’m missing something.

We addressed the reviewer´ comment in the previous version of the manuscript considering the CPUE in the analysis. In fact, the CPUE comprehend the fishing effort, therefore the final results already include this variable.
